# A Review of the Impact That Healthcare Risk Waste Treatment Technologies Have on the Environment

**DOI:** 10.3390/ijerph191911967

**Published:** 2022-09-22

**Authors:** Thobile Zikhathile, Harrison Atagana, Joseph Bwapwa, David Sawtell

**Affiliations:** 1Faculty of Natural Sciences, Mangosuthu University of Technology, 511 Griffiths Mxenge Highway, Umlazi, Durban 4031, South Africa; 2Institute for Nanotechnology and Water Sustainability, University of South Africa, Pretoria 0003, South Africa; 3Faculty of Engineering, Mangosuthu University of Technology, 511 Griffiths Mxenge Highway, Umlazi, Durban 4031, South Africa; 4Department of Engineering, Manchester Metropolitan University, John Dalton Building, Chester Street, Manchester M1 5GD, UK

**Keywords:** health-care risk waste, treatment technologies, climate change, healthcare, environment, health

## Abstract

Health-Care Risk Waste (HCRW) treatment protects the environment and lives. HCRW is waste from patient diagnostics, immunization, surgery, and therapy. HCRW must be treated before disposal since it pollutes, spreads illnesses, and causes harm. However, waste treatment increases the healthcare sector’s carbon footprint, making the healthcare sector a major contributor to anthropogenic climate change. This is because treating HCRW pollutes the environment and requires a lot of energy. Treating HCRW is crucial, but its risks are not well-studied. Unintentionally, treating HCRW leads to climate change. Due to frequent climate-related disasters, present climate-change mitigation strategies are insufficient. All sectors, including healthcare, must act to mitigate and prevent future harms. Healthcare can reduce its carbon footprint to help the environment. All contributing elements must be investigated because healthcare facilities contribute to climate change. We start by evaluating the environmental impact of different HCRW treatment technologies and suggesting strategies to make treatments more sustainable, cost-effective, and reliable to lower the carbon footprint.

## 1. Introduction

Global climate change is no longer an alarming future danger, but a dawning reality that is already causing unsettling changes in the natural and human surroundings and destroying the delicate balance of our planet’s ecosystem and the species that rely on it [1,2,3,4]. Climate change is defined as a long-term major shift in weather patterns [5,6]. A changing climate has an impact on human health since human health is inextricably linked to environmental health [7,8,9,10,11]. The World Health Organization continues by stating that a healthy planet with access to clean drinking water, enough food, safe housing, and favorable social conditions is necessary for long-term health [12]. However, the changing climate affects all of these.

There are several types of facility in the healthcare sector, which are commonly referred to as healthcare centers or medical centers. They refer to a location where medical advice, diagnosis, or treatment is provided [13,14]. Healthcare facilities produce a lot of HCRW as a byproduct of providing care and medication [15,16]. The generated HCRW must be treated before disposal because it can cause illness as well as environmental contamination and damage if left untreated [16,17]. The most common methods of treatment are autoclaving, microwaving, or incineration [18]. These prevalent treatment procedures are high-energy procedures [19]. The other treatment methods are chemical methods and plasma pyrolysis [20]. However, HCRW treatment and treatment technologies face obstacles. They have substantial harmful environmental consequences. Because their operation requires high temperatures, they consume an enormous amount of energy. Moreover, during incineration, incinerators produce fly ash, bottom ash, and fugitive gases, such as vapors or particles [21,22,23]. The wastes produced by electrostatic precipitators and bag filters comprise the fly ash. The amount of fly ash produced is estimated to be 3–10% of incineration waste. The heavy metal concentrations in fly ash are higher than those in bottom ash [24].

The high expense of procuring and operating the technology adds to the challenges caused by the treatment methods. Some countries cannot afford to treat HCRW. The waste is mixed up with household waste and disposed of in landfills [16,25,26].

Leachate and methane gas are produced at landfill sites as a result of both treated and untreated HCRW [18,27,28,29]. Leachate is a very hazardous liquid generated in landfills from high-water-content waste [29], and methane is a greenhouse gas that is the leading contributor to climate change [30,31].

There has been limited research on the impact of HCRW treatment technologies on climate change. However, there is strong evidence that healthcare institutions, with their large carbon footprint, contribute to anthropogenic climate change. The management of HCRW has a crucial role in this contribution [32,33]. Consequently, the carbon footprint of healthcare institutions jeopardizes the health of the communities that these facilities are intended to serve [33]. For the healthcare industry to uphold its commitment to promoting health, it must take reasonable measures to protect human health and the environment by reducing its carbon footprint, which contributes to climate change [34,35]. The greatest implication of climate change is the potentially devastating impact it has on human health [36,37,38]. This can be accomplished, in part, by reevaluating the current waste management practices.

Current research on HCRW focuses mostly on HCRW composition, treatment technologies, and the health and environmental impact of HCRW. There is a shortage of knowledge regarding the carbon footprint of current treatment systems. This research will contribute to the existing body of knowledge and provide a theoretical foundation for the development of future interventions pertaining to HCRW management practices and behavior.

This review offers an overview of the existing HCRW management system’s environmental impact. It further highlights how the carbon footprint of the existing management techniques contributes to climate change. The review recommends alternative strategies for the treatment and management of HCRW, which will contribute to the decision-making process. It also aids the development, selection, and planning of future environmentally sustainable hospital waste-management systems.

This work is based on a review of the relevant literature and investigates how the scientific community has raised the discussed issues. ‘Elsevier,’ ‘Science Direct,’ ‘Scopus,’ and ‘Google search engines’ were used to study the scholarly literature on HCRW, HCRW treatment technologies, recycling, and the carbon footprint of healthcare facilities. Over a hundred documents were collected and properly analyzed. The inclusion criteria included peer-reviewed journal papers and conference proceedings, legitimate book chapters, published and unpublished reports, and selected web references.

## 2. Discussion and Analysis Regarding Health Care

### 2.1. Health Care Risk Waste

The definition of HCRW differs depending on the category and activity of the waste generated. It is waste generated from diagnostic, monitoring, preventive, curative, or palliative activities in the fields of veterinary and human medicine, and it can be further defined as “any solid or liquid waste generated in the diagnosis, treatment, or immunization of human beings or animals, in research pertaining thereto, or in the production or testing of biologicals or living organisms” [39]. HCRW is produced by a variety of facilities. According to the World Health Organization, HCRW is defined by the waste that they generate. HCRW “includes all medical waste created within healthcare facilities, research centers, and laboratories.” Furthermore, it includes the same sorts of waste originating from minor and dispersed sources, such as waste generated during home healthcare (e.g., home dialysis, self-administration of insulin, recovery care), as illustrated in Figure 1 and Table 1. Furthermore, the waste generated is divided into many types. Waste classification is critical for waste handling and treatment. As stated in Table 1, waste is categorized as infectious waste, pathological waste, sharps’ waste, chemical waste, pharmaceutical waste, cytotoxic waste, and radioactive waste [39,40].

HCRW contains highly toxic metals, chemicals, pathogenic viruses, and bacteria that have the potential to cause illness and disease [41]. The diseases range from gastro-enteric, respiratory, skin infections, Hepatitis B and C (Jaundice), and HIV/AIDS [42,43]. The diseases can be transmitted directly or indirectly through contaminated air, soil, groundwater, and surface water [44]. The waste also has the ability to pollute and damage the environment. The wind can transport bacteria and harmful elements from carelessly discarded HCRW to people, causing disease. Domestic animals may graze in areas containing HCRW if the waste is disposed of in fields that are accessible to them. The waste microbes can be reintroduced to humans through the food chain [42].

**Figure 1 ijerph-19-11967-f001:**
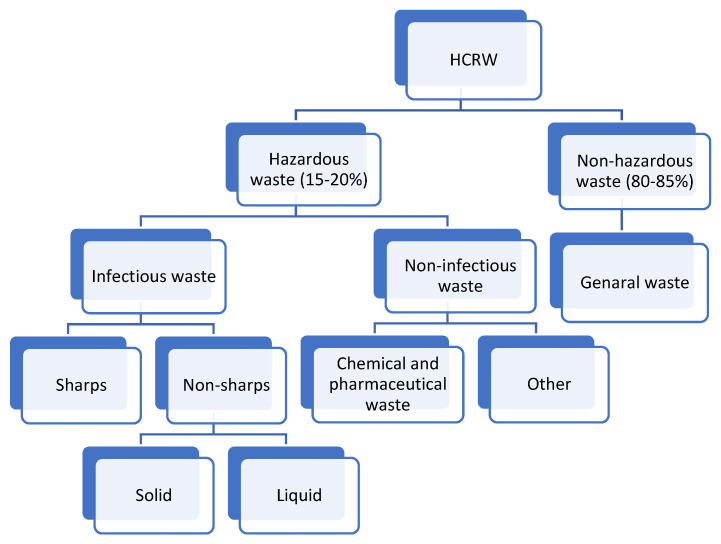
HCRW categories [42].

There are a variety of HCRW classifications based on their origins. It is essential to note that, just as HCRW is different, so are its treatment approaches. In addition, the used treatment procedures depend on a range of factors, such as the availability of the technology, the financial repercussions, and the public’s approval. Table 1 illustrates how the treatment technique is selected based on the type of waste being processed, the availability of resources, and the ease of access.

**Table 1 ijerph-19-11967-t001:** HCRW generators and management.

HCRW	Description	Sources	Management
Infectious waste	Waste contaminated with blood and other bodily fluids Cultures and stocks of infectious agents from laboratory work Waste from patients with infections	Hospitals Laboratories Research centers Mortuary Autopsy centers Blood banks Nursing homes Home health care Acupuncturist Paramedic and ambulance services Animal research Physicians’ offices Dental clinics Chiropractors Psychiatric hospitals Cosmetic piercing and tattooing Institutions for disabled people	Non-Burn Thermal Technologies Autoclaves Hybrid Steam Systems Microwave Units Frictional Heat Treatment Dry Heat Systems Chemical Technologies Alkaline Hydrolysis Incineration [40,45]
Pathological waste	Human tissues Organs or fluids Body parts Contaminated animal carcasses	Hospitals Laboratories Research centers Mortuary Autopsy centers Paramedic and ambulance services Animal research	Incineration [39,40]
Sharps waste	Syringes Needles Disposable scalpels Blades Razors Broken and/or contaminated glass Microscope slides Certain medical saws or amputation equipments Knives	Hospitals Laboratories Research centers Autopsy centers Blood banks Nursing homes Home health care Acupuncturist Paramedic and ambulance services Animal research Physicians’ offices Dental clinics Chiropractors Psychiatric hospitals Cosmetic piercing and tattooing Institutions for disabled people	Non-Burn Thermal Technologies Autoclaves Hybrid Steam Systems Microwave Units Frictional Heat Treatment Dry Heat Systems Chemical Technologies Alkaline Hydrolysis Incineration [39,45]
Chemical waste	Solvents Reagents used for laboratory preparations Disinfectants Sterilant Heavy metals contained in medical devices (e.g., mercury in broken thermometers) Batteries	Hospitals Laboratories Research centers Mortuary Autopsy centers Blood banks Nursing homes Home health care Acupuncturist Paramedic and ambulance services Animal research Physicians’ offices Dental clinics Chiropractors Psychiatric hospitals Cosmetic piercing and tattooing Institutions for disabled people	Ion exchange Precipitation Oxidation and Reduction Neutralization [40,46]
Pharmaceutical waste	Expired, unused and contaminated drugs Vaccines	Hospitals Laboratories Nursing homes Home health care Physicians’ offices Dental clinics Psychiatric hospitals Institutions for disabled people	Non-Burn Thermal Technologies Autoclaves Hybrid Steam Systems Microwave Units Frictional Heat Treatment Dry Heat Systems Chemical Technologies Alkaline Hydrolysis Incineration [46]
Cytotoxic waste	Waste containing substances with genotoxic properties (i.e., highly hazardous substances that are mutagenic, teratogenic, or carcinogenic), such as cytotoxic drugs used in cancer treatment and their metabolites	Hospitals Research centers Nursing homes Home health care Animal research	Incineration [39]
Radioactive waste	Such as products contaminated by radionuclides including radioactive diagnostic material or radiotherapeutic materials	Hospitals Research facilities	Most radioactive waste requires packaging in specially engineered containers for safe storage and disposal [40]

### 2.2. Generation

Accurate figures for the generated HCRW are not known but research suggests that the amount of generated waste is enormous [47,48]. This is due to a variety of variables, such as population growth, an increase in chronic diseases, pandemics, excessive usage of disposable things, and so on [45,49]. It is estimated that South Africa generates approximately 42,200 tons of HCRW a year [50], and there is an annual increase of 1.5% in the generation [50]. The HCRW is generated when medical care is provided by healthcare facilities [41]. Hospitals, clinics, community health centers, laboratories, research institutions, dental facilities, emergency services, ports of entry, veterinarian practices, nursing homes, and forensic pathology services are among the generators [51,52]. Hospitals are major HCRW generators, with waste coming from general wards, acute care wards, injury units, theatres, medical laboratories, accident and emergency, admin, and support offices. The waste created is made up of 85% general HCRW (non-infectious waste), 10% hazardous waste (infectious waste), and 5% chemical/radioactive waste (hazardous waste) (Figure 1) [42].

The creation of HCRW is contingent on the services provided (Table 1). Healthcare institutions in developing nations, including the Middle East, South America, and India, create between 1.0 and 3.0 kg of HCRW per person per day [42], whereas the United Kingdom generates 5 kg of HCRW per person [44].

Instead of employing reusable items, healthcare facilities frequently use disposable ones [45]. During the COVID epidemic, there was a significant surge in the production of HCRW, notably disposable goods such as gloves and masks [46].

### 2.3. Management

HCRW management is managing waste in a manner that prevents the spread of diseases and protects the environment. It is collecting, transporting, treating, and disposing of all HCRW materials. HCRW management is further elaborated as managing HCRW from the initial handling, collecting, transporting, treating, disposing, and monitoring of waste materials and reducing the amount and hazards of waste [53,54].

HCRW is highly infectious and hazardous [54]. It has been established that the mismanagement of HCRW has detrimental effects on health and the environment, making proper HCRW management a top priority [42]. HCRW has to be handled in a manner that protects people from exposure to infection by disease-carrying microorganisms [42], and in a manner that does not cause environmental degradation. As a result, in many developed countries, specific laws and regulations have been formulated and implemented for proper HCRW management [55]. However, the situation is quite different in developing countries. Law and regulations have been formulated but not fully implemented [55]. Developed countries such as the United Kingdom are complying with HCRW guidelines with regard to HCRW management [56]. In South Africa, the management of HCRW is highly regulated [50]. However, in some developing countries such as Nigeria, and Ethiopia, there is poor management of HCRW [50,54,57].

It is important that HCRW should be managed separately from general waste. For the optimal management of HCRW, all of the stages of the medical product’s life cycle should be considered [58]. The management of the waste is a process and it starts with waste generation where HCRW is generated. It is then followed by waste segregation using correct containers. The containers must be labelled properly. It is then collected from the point where it was generated to the temporal storage. The waste is then transported by an approved and compliant vehicle to a licensed treatment facility for waste to be treated. The final process after treatment is final disposal [54].

It has been established that all of the generated HCRW must be treated before final disposal. The main objective of treating the waste is to make HCRW safe and this applies to infectious waste, anatomical waste, all clinical waste, and medical waste [59]. The treatment methods used to treat the waste are high and low-heat treatment systems. The high heat is the use of incinerators, and the low heat is the use of autoclaving, microwave irradiation, chemical methods, and plasma pyrolysis. However, incineration and autoclaving are the most common treatment methods of treatment in a majority of countries, including South Africa [60,61,62].

#### 2.3.1. Incineration

HCRW incineration is a high-temperature dry oxidation process that converts waste to residual ash and gases. The incineration consists of a primary combustion chamber that operates at between 800 and 1000 °C and a secondary chamber operating at between 850 and 1100 °C [42]. It destroys microorganisms that are in the waste but only if the incinerator is operated properly. However not all HCRW can be incinerated. Only human anatomical waste, such as human tissues, organs, body parts, and animal carcasses are incinerated. Pharmaceutical waste in any form or container; microbiological cultures; cytotoxic and cytostatic-contaminated waste; contaminated metal parts; wastes from chemotherapy treatment; mercury, volatile and semi-volatile organic compounds and radioactive wastes are not incinerated. In addition, pressurized gas containers; large amounts of reactive chemical; silver salts, photographic or radiographic chemicals; halogenated plastics such as polyvinyl chloride (PVC); mercury and cadmium compounds; and sealed ampoules or ampoules containing heavy metals cannot be incinerated [39,63,64].

Research shows that HCRW incineration is not the best solution for waste disposal contrary to popular belief. The process of waste incineration pollutes the environment and causes ill-health [65]. Furthermore, it is the law of conservation of matter. This law states that no waste can be completely destroyed, no matter the characteristic of the waste. The waste that is incinerated does not disappear, however, it is transformed into different physical phases [66,67]. In the process of incineration, exhaust gases are released in the atmosphere, ash is generated at the bottom of the incinerator, and fly ash is trapped in the incinerator filters. Based on research, all of the incinerators emit three types of toxic pollutants to the environment even if the emissions’ control technologies are applied. These toxic pollutants are heavy metals, partially combusted chemicals, and new chemicals that are synthesized during the incineration process in the combustion chambers. The end product of the incineration process, the bottom ash, contains toxic end-products of the combustion. The bottom ash is landfilled, thereby contaminating the environment in the disposal site and groundwater [66,67,68,69].

Some of the hospital materials, such as intermittent catheters, are made of rubber or polyvinyl chloride (PVC), making them softer and more flexible [70], and syringes have components of polyvinyl chloride (PVC) [68]. The incineration of HCRW that contains polyvinyl chloride materials results in the formation of acidic gases, such as sulfur dioxide and hydrochloric acid.

Some of the healthcare risks, such as pharmaceutical waste, require high temperature of approximately 1250 °C for about 2 s for the complete destruction of the polychlorinated biphenyls (PCBs) [42]. If the temperature is low or time is short, new molecules are synthesized in the incinerator. An example of this is when polychlorinated biphenyls (PCBs) are incompletely combusted, then dioxins and furans, such as polychlorinated dibenzo-dioxins and polychlorinated dibenzo-furans, are generated as different waste. These are extremely toxic, far more so than the chlorinated biphenyls (PCBs) and their precursors. These dioxins and furans did not exist in the medical waste prior to incineration [71].

The incineration of HCRW seems to be the most preferred method of treating waste because of its success in eliminating micro-organisms and the fact that it reduces the quantities of waste [62,72]. The high success rate of eliminating micro-organisms is because the temperature that is used is extremely high, but this means more energy and fuels are used [42]. In addition, the incineration process must always be completed according to the time schedule of the incinerator [66]. Some of the microorganisms can survive if the incineration process is not complete. Furthermore, not all of the adverse effects of incinerators have been fully considered and well-studied. In addition, the growth in the generation rate of HCRW increases the use of the incinerators thereby increasing the environmental and health impact of the incinerators. Considering this information, it is therefore important to consider the use of incinerators as a temporary solution and it should be supplemented with another solution that does not endanger the environment and human health.

#### 2.3.2. Autoclaving

The autoclaves are used to treat HCRW before the waste is disposed of in the landfill sites. During the process of waste segregation, the generators of HCRW place microbiological and biotechnological waste in designated containers. It is these containers that are placed in autoclaves in batches or cycles. The autoclave is a metal cylindrical vessel fitted with a steam jacket. Autoclaves are heated up to 121 °C for about 30 min and steam gets added into the process to maintain the required temperature. The purpose of the steam jacket is to reduce condensation in the vessels thereby preventing heat loss [73]. The steam is supplied into the system via boilers and electricity is used to heat the boilers [73].

HCRW treatment is important in order to prevent infection and environmental pollution. However, the challenge is that generally, the currently used technologies have their own complications. However, the autoclave presents fewer problems compared to the other treatment technologies. This has made the autoclave method to be better preferred [62]. This is because autoclaves are cheaper to procure, use less energy, and have less environmental pollution compared to the incinerators. However, the preference is based on the development of the country choosing the technology. Their use is still very expensive for developing countries.

#### 2.3.3. Microwave Irradiation

The research suggests that this is a very effective method of treating HCRW. Microwaves are electromagnetic waves that have frequencies that are falling below the range for infrared waves but above the ultra-high frequency. It works by converting electrical energy to microwave energy. The microwave energy is then used to generate steam from the moisture that is in the HCRW to be treated [73]. Some microwave methods apply low-frequency radio waves that inactivate all of the microorganisms that are in the HCRW [63]. The drawback with this method is that it is a very costly method, and it is therefore not so popular, especially in developing countries [74]. Furthermore, the use of steam results in the treated HCRW contributing to the substantial amount of leachate generation in the disposal site thereby contaminating ground water and contributing to generating methane gas [29,31].

#### 2.3.4. Chemical Method

Chemicals were used for disinfecting medical equipment, hospital walls, and floors. However, it is now also used for treating HCRW [75]. This is a form of treatment where sodium hypochlorite (NaClO, 5%) hydrogen peroxide (H_2_O_2_, 30%), and Fenton reagent (FeCl_2_·2H_2_O; 0.3 g in 10 mL H_2_O_2_, 30%) is used to disinfect contaminated HCRW [76]. The challenge with this treatment technology is that it does not sterilize but it disinfects. Furthermore, it is generally used to disinfect liquid waste such as blood, urine, human waste, or hospital sewage. Therefore, the HCRW that can be treated by this method is limited.

#### 2.3.5. Plasma Pyrolysis

Plasma pyrolysis is the technology used for the safe treatment of HCRW. It converts organic waste into useful by-products that can be used for commercial purposes. Plasma generates intense heat that disinfects all of the microorganisms in the waste. When the HCRW comes into contact with plasma-arc, it is pyrolyzed into CO, H_2_, and hydrocarbons. These gasses are burned and produce a high temperature of approximately 1200 °C [18].

### 2.4. Environmental Impact of HCRW Treatment

Because HCRW is infectious, it contains harmful microbes, hazardous chemicals, and radiation. Therefore, treatment is required prior to disposal. However, HCRW treatment technologies have negative environmental effects. Due to the massive greenhouse gas emissions produced, the usage of incinerators to handle HCRW has sparked contentious debates [32]. Incinerators make persistent organic pollutants (POPs) such as dioxins and furans (PCDD/F), polychlorinated biphenyls (PCBs), polycyclic aromatic hydrocarbons (PAHs), chlorobenzenes (CBz), etc. [77]. Although the exact quantity of greenhouse gases emitted by incinerators is unknown, it is considered to be substantial [13,21].

The incineration of HCRW creates polluting gases and fly ash, bottom ash, and scrubber water filters. Furthermore, incineration generates fugitive gases, such as vapors or particles that escape during the combustion process. The fugitive gases are emitted not only during incineration but also from bottom ash and fly ash hoppers during the transfer of bottom ash and fly ash from transport vehicles to landfill sites [77].

Waste incineration is problematic since it contributes to air pollution, and contaminants bioaccumulate within organisms [69]. The pollutants exponentially increase as they ascend the food chain due to bioaccumulation [71]. Because of this, species such as people, animals, and plants absorb contaminants more quickly than their systems can eliminate them.

Greater quantities of polychlorinated dibenzo-para-dioxins (PCDDs) and polychlorinated dibenzofurans (PCDFs) were found in chicken meat and eggs than in the soil in which the hens foraged [78]. Cows also exhibit the same bioaccumulation [79]. Other research found that the concentration of dioxins and furans in cows’ milk from cows grazing near incinerators was higher than in cows grazing further away from incinerators [80]. Another study found higher dioxin levels in the blood of Korean citizens living within a 5-km radius of an industrial waste incinerator [81].

In addition to bioaccumulation, the air contaminants stay in the atmosphere for an extended period of time [69]. Air pollutants, such as carbon monoxide (CO), nitrogen dioxide (NO_2_), ozone (O_3_), sulfur dioxide (SO_2_), and fine particulate matter (PM2.5), have been accumulating over South Africa’s Highveld [82].

In addition, the treatment methods are energy intensive. They function at extremely high temperatures. The operators are guided by the duration for which the trash will be processed. The HCRW treatment technology utilizes enormous amounts of energy or fuel. An incineration temperature is between 800 and 1000 degrees Celsius. Other treatment methods are heated to temperatures exceeding 146 °C [83]. The electricity required to autoclave 1 kg of HCRW ranges between 0.2 to 1.4 kWh kg [19]. An autoclave that handles 15 kgs of HCRW requires 180 kWh of energy and 5400 L of water every day. The latter electricity consumption is computed depending on the technology’s operation and idle time. The consumption level is equal to the amount of power and water consumed by ten families [19].

The most-used HCRW technologies do not completely eliminate waste. Through the process of incineration, the processed waste may be shrunk [84]. Generally, autoclaves are equipped with shredders so that the treated waste can be shredded and then compacted to reduce its volume. The incinerator residues and burned waste must still be disposed of at a landfill [30,84]. This contributes to the environmental issues associated with waste landfilling, such as the creation of leachate and the production of methane gas [29,31]. The HCRW management technologies unintentionally create the most severe environmental contamination, which contributes to anthropogenic climate change. The impacts of climate change on human health and the environment cannot be refuted [85]. The health repercussions of climate change are becoming increasingly apparent and frequent [86,87]. Direct repercussions include heat stress and fires, flooding, and storms [88]. Malnutrition, due to agricultural failure, and changing infectious disease patterns are secondary effects [86,89,90,91]. The healthcare industry has a substantial environmental impact because it is a very energy-intensive industry. It is the fifth-largest greenhouse gas (GHG) emitter on the planet, accounting for 4.4% of worldwide net emissions [41,88,92]. The mismanagement of waste is regarded as one of the energy-wasting activities in the healthcare industry [92].

## 3. Discussion on Challenges and Analysis on Prospective or Relevant Technology Options for the Treatment of HCRW

Table 2 depicts the optimal strategy for treating various types of HCRW. It describes treatment techniques as well as the benefits and drawbacks of HCRW therapy technology. Nevertheless, the downsides of each therapeutic method exceed their positives. This is because the treatment methods require natural resources to operate. The table gives more information on the characteristics of various technologies, the treatment temperature, the type of waste, and the time required to treat HCRW.

Evidently, numerous creative choices or solutions for HCRW treatment are being evaluated and implemented. Some of them may not be scalable in the healthcare industry due to their energy consumption, operating, or capital expenditure expenses. The HCRW management technologies unintentionally contribute to anthropogenic climate change, which is the most serious kind of environmental pollution, due to their energy consumption and management practices. In addition, the contribution of treatment technologies and management systems to climate change substantially increases the carbon footprint of healthcare facilities.

There is a lack of data on the true costs of HCRW disposal due to commercial sensitivity. The cost of disposal is expected to be GBP 0.45/kg in the United Kingdom and USD 0.79/kg per ton in the United States [93]. As a result, developing countries struggle to treat waste. This is due, in part, to the fact that waste-treatment technologies are heavily influenced by a country’s level of development in terms of cost and accessibility.

The bulk of choices for waste disposal and treatment are developed by a healthcare facility or group of hospitals [92]. Even though involvement in HCRW treatment programs is intended to have a significant impact, the majority of settings lack the fundamental knowledge of HCRW solutions and do not even practice basic waste segregation. This is mainly prevalent in developing nations when there is a lack of appropriate legislation, robust follow-up mechanisms, and policies to support actions pertinent to the effective management of HCRW. While there are a number of ways for treating or disposing of HCRW, including autoclaving, incineration, microwave, reverse polymerization, chemical disinfection, and pyrolysis, the most common are autoclaving, incineration, and microwave. Occasionally, the acquisition of expensive technology-related equipment impedes the usage of technology. In addition, the essential education, training, and waste segregation knowledge may look insufficient, especially in developing nations where such resources are few.

Numerous methods, such as incineration, are considered safe when used effectively; nevertheless, strict adherence to norms and laws is required to ensure such safety, and such adherence is not always enforced. Despite the fact that many developed nations are making small, deliberate changes, such as improved resource decision-making tools, better, more appropriate use of existing resources, and participation in improvement initiatives such as recycling, reuse, and reprocessing methods’ initiatives, these changes must be encouraged and implemented on a larger scale. The current trend in technology is the development of zero-waste, energy-efficient devices. Currently, the trend toward adopting renewable energy is increasing. In addition, integrated processes occurring within a single unit are considered economically viable. They have the benefit of requiring less space and being energy efficient, but the operational and capital expenditures are still a matter of contention. Once it is effectively included into the treatment of HCRW while consuming less energy, the zero-waste strategy may be a viable option.

Typically, technological breakthroughs are minor, and it may be difficult to globally adopt a large technological advancement. Notably, even if such education is provided and a global understanding of HCRW disposal develops, many issues pertaining to HCRW disposal, including the pollution, global warming, and global health repercussions, remain unsolved. Particularly in developing nations, there is an urgent need for a full overhaul and reform of the systems utilized for the disposal of HCRW. Open-pit burning of hazardous and non-hazardous waste is still an issue in many parts of the world. Massive quantities of toxic and unpleasant gases are released into the atmosphere because there has been no preceding inertization.

In addition, there is still a significant knowledge gap between the developed and developing nations regarding the treatment and disposal of HCRW; however, leveling the playing field does not eliminate the challenges related to HCRW disposal. The existing scalable, practical, and realistic techniques of HCRW disposal have a multitude of challenges, mostly due to a lack of necessary resources and noncompliance with guidelines and legislation, many of which cause long-term damage to the environment and, consequently, global human health. While there are initiatives and programs in place to reduce the quantity of waste that must be discarded and educational programs to better equip some regions of the world with new habits, these gains remain limited.

As a result, when implemented on a big scale, even the most advanced solutions to global waste pose long-term harm to the environment and human health. Furthermore, while public attention is routinely drawn to waste reduction at the home and commercial levels, there is rarely explicit attention drawn to HCRW. This is a problem, given HCRW’s large contribution to global garbage. Although several pieces of research have proposed potential solutions to the aforementioned HCRW management difficulties, only a few implementation, feasibility, and follow-up studies have been completed.

In a similar vein, few pieces of research clearly correlate certain facilities to the start or worsening of a chronic disease, even though numerous studies draw comparisons between HCRW’s contribution to greenhouse gas emissions. Similar research on infectious diseases are frequently conducted in underdeveloped nations and, to a lesser extent, in wealthy nations. In order to comprehend the precise risks posed by different HCRW techniques and the diseases they have directly touched, further in-depth, extensive investigations should be undertaken in the future. An overview of the various HCRW management techniques currently in use is given in this review. It advances the state of current information by looking into both the effectiveness of present methods as well as brand-new, more experimental HCRW therapy options and highlighting the variations between poor and rich countries’ HCRW disposal strategies in terms of their effects on world health. The review does not go into detail regarding the precise processes of operation of each of the approaches used, which is a shortcoming of this study. The review is further constrained in its investigation of the subject due to the paucity of literature in this area, which is reflected in the severe absence of precise data on the direct, explicit consequences that HCRW has on human health. Preliminary findings suggest that present HCRW techniques are insufficient and can have a detrimental influence on the environment by increasing the carbon footprint, hence there is a need for additional in-depth, extensive investigations on this topic.

### Suggested Steps, Operations or Actions to Be Taken for HCRW

Figure 2 depicts numerous components that can be involved in efficiently dealing with HCRW. There are numerous levels at which HCRW should be explored. Among them are the management and strategic levels, practical and site levels, and administrative, financial, and human resource levels. All of the components can contribute to the development of an effective strategy for the better management of HCRW. It is suggested that they be mixed to produce the best outcomes.

## 4. Conclusions

The generation of HCRW is rising at an alarming rate, and the waste that is being produced must be handled since HCRW that has not been treated can have a substantial impact on both human health and the environment. In addition, there is not a single waste-treatment method that is now available for healthcare hazards that takes into account all of the impact on the environment. Furthermore, it is clear that the treatment of HCRW has other unexpected side effects, including the degradation of the environment and the contribution to climate change. It is imperative that immediate attention be paid to the role that HCRW treatment methods play in climate change. Technology that safeguards both human health and the natural environment must be utilized for the treatment and management of HCRW.

The existing contribution of HCRW to climate change calls for immediate attention, and the systems that make use of renewable ways call for significant consideration. This is crucial in light of the destructive effects that climate change has on human health. When proper and alternative HCRW treatment technologies are utilized, the ultimate result will be systems that are better for the environment and more cost-effective. In addition, the utilization of innovative solar treatment methods will result in the production of apparatus that is both dependable and effective.

It is necessary to develop a new method for the treatment of HCRW that is able to significantly cut down on the amount of energy that is consumed during the treatment process and cut down on the amount of HCRW that is sent to landfills by producing useful end-products in order to reduce the amount of waste that is sent to landfills and n the amount of environmental pollution that is caused by landfills.

The challenges resulting from the current HCRW treatment technologies impair the health of the communities that healthcare facilities are intended to serve. In order for the healthcare industry to uphold its obligation to provide health, it must take reasonable precautions to protect human health and the environment. The healthcare industry must take fundamental steps to employ technology that does not rely on natural resources. One method to accomplish this is by re-examining the present waste management systems and evaluating HCRW treatment technologies that utilize alternative energy sources, such as solar energy.

## Figures and Tables

**Figure 2 ijerph-19-11967-f002:**
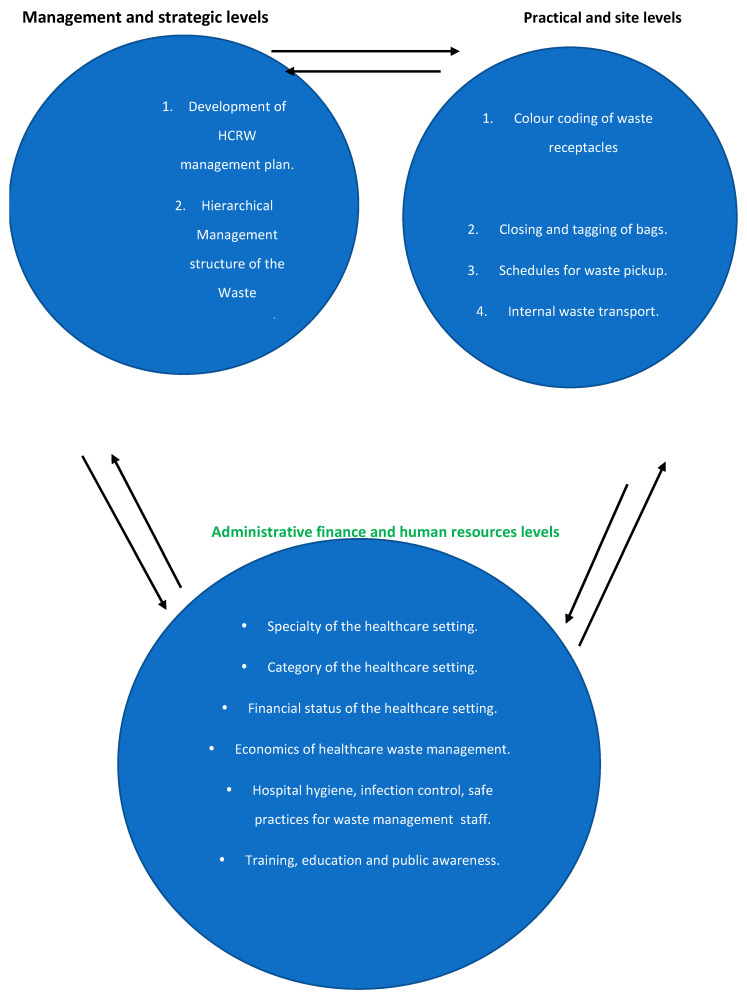
Components of levels for an effective HRCW management.

**Table 2 ijerph-19-11967-t002:** HCRW treatment technologies [64].

Treatment Methods	Description	Types of Waste Treated	Temperature for Treatment	Treatment Time	Advantages	Disadvantages
Incineration	High-heat treatment converts waste into ash and exhaust gases.	Anatomical, infectious, and pharmaceutical wastes	Primary chamber 800–900 °C Secondary chamber 900–1200 °C	4–6 h per batch	Suitable for treatment of all types of hazardous HCRW. Reduces the volume significantly.	Produces air pollutants, carcinogens (dioxins, polychlorinated biphenyls, polycyclic aromatic compounds) and harmful gases (HCl, HF, SO_2_). Very costly.
Autoclave	Use saturated steam to disinfect infectious waste.	Infectious, pharmaceutical, and sharps	121–140 °C	30–60 min per batch	Has better public acceptance than incinerators.	The disinfected waste is landfilled. The discharged moisture contaminates the environment.
Microwave	Steam-based technology. Uses microwave disinfection to treat waste.	Infectious, pharmaceutical, and sharps	95–100 °C	≥30 min per batch	Has better public acceptance than incinerators.	The disinfected waste is landfilled.
Reverse polymerization	Uses microwave energy to treat waste. Shredding is applied to the final sterilized carbon residue.	Infectious waste	180–370 °C	50–80 min per batch	Decreases the quantities of waste.	Use of Sodium Hydroxide (NaOH) and a scrubber to control gaseous emissions. Production of wastewater. Extremely costly.
Chemical disinfection	Uses a chemical technology with sodium hypochlorite as a disinfectant.	Liquid waste	95–155 °C	25 min exposure per batch	Low air emissions. Simple and convenient, good deodorization effect.	Production of liquid waste containing sodium hypochlorite (NaOCl).
Pyrolysis	Heats waste organic components under oxygen-free or -depleted conditions, breaks chemical bonds to transform combustible liquid and gas.	Infectious waste	540–830 °C	45 min	Pyrolysis technology has a high energy recovery rate, minimal secondary pollution, and sufficient economics.	Commonly used for organic materials. It occurs at high temperatures.
Gasification	Gas cloud formed by the ionization of an inert gas, usually referred to as the fourth state of matter.	Infectious waste	3000 °C	1/1000 s	Treats all HCRW.	Expensive. Energy demanding.

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
