# Peer review of "A Review of the Impact That Healthcare Risk Waste Treatment Technologies Have on the Environment"

_ijerph, 2022, doi:10.3390/ijerph191911967_

Round 1

Reviewer 1 Report

After going through the manuscript "A Review of the Impact of Health Care Risk Waste Treatment 2 Technologies Have on the Environment ", I would give my comments below.

1.I think there are parallel work with some new review papers that publish in recent months such as:

a. hhttps://doi.org/10.3390/healthcare9030284

b. https://doi.org/10.1177/0734242X19857470

c. https://www.researchgate.net/profile/Y-Babanyara/publication/285242270_Poor_Medical_Waste_Management_MWM_practices_and_its_risks_to_human_health_and_the_environment_a_literature_review/links/56660a6e08ae4931cd626842/Poor-Medical-Waste-Management-MWM-practices-and-its-risks-to-human-health-and-the-environment-a-literature-review.pdf

The scope covered can be considered as not a new review manuscript for the present, so, what makes this review different from the others and from the most recent ones?

2. Abstract:

2i.It is suggested to add the key points on how poor management of HCRW can affect human health and  caused climate change.

2ii. The words ‘human health and environment were repeated few times. Rephrasing required. (Line 19-22).

Overall, the point addressed in this section must be written concisely.

3. Section 3.1:

3i. It is suggested to reproduce the fig 1 to develop a better graph.

3ii. Table 1: The author should add source (citation) in each row of final column of the table.

4. Section 3.2:

4i.This section is not well discussed. And the figures shown require major improvement. For example fig 2 – my suggestion is to generate your own data and construct new pie chart. In a review journal, recent data is required, so, should use the recent numbers.

4ii.Fig 3: The reference is too old. This fig looks very poor in term of presentation

4iii, Should provide a comprehensive part between all of the treatment of HCRW in the experimental and field-scale till now. . Add another table  or figure.

4iv.A review paper not only should summarize recently published works, but also should contain critical and comprehensive discussions. Therefore, check writing for the whole manuscript. The review should not only presented by listing what have done by others.

5. Section 4.1 : Figure require major improvement. . In objective (page 1), the author mentioned about cost. But, there is no discussion about cost. My suggestion is to discuss about cost benefit and relate with the carbon footprint .

5i. Section 4.1 : could increase quality of the manuscript. Section 4 and 4.1

6. Conclusion is not well addressed and tallied with objectives.

Author Response

Comment 1: Unlike the other reviews, this one reveals that the healthcare sector continues to have a significant carbon footprint. HCRW treatment technologies are a major contributor to this problem. This research also demonstrates how the healthcare industry is unintentionally contributing to a rise in the number of people who require medical attention.

Section 2i: Line 18-19

Section 2ii: Changed

Section 3i: Graph removed because of the comments of the second reviewer.

Section 3ii: Citations included at the end of each column

Section 4i: figure 2 was removed and figure 3 changed

Section 4ii: reference changed

Section 4iii - Section 5: improvements were made, line 354 - 364.

Section 6: 462 - 468 and line 474 - 481

Reviewer 2 Report

The manuscript has a 30% similarity index, using turnitin.com. I suppose the authors could bring this further below 10%.

0 -  The manuscript should be revised for grammar. A number of tautologies and syntax errors noted. I suggest the authors use grammarly.com or quillbot.com

14 – Define HCRW before discussing its usefulness

15 - However, the treatment of the waste increases the carbon footprint of the healthcare sector resulting…… Briefly explain how

26 – rather than state the aim of the paper, capture the summary of salient issues raised in a line or two. This will help to provide the link necessary to justify the conclusion in the Abstract.

31 – Increase keywords by 2 at the least

90 – 97 – This section is not necessary. Delete. The article is a review and should be presented as such. The introduction of a Materials and Methods sections presents the article as though it is metadata analysis.

98 – Remove this subheading. Just let the entire review flow.

131 – Left-justify Table 1. Do same to all other Tables

444 – Provide reference for Figure 4, else state if it is modified or the intellectual property of the authors.

Author Response

Quillbot.com was used to improve grammar

 Line 14 - 17, HCRW was defined

Line 21 -22, Brief explanation

Line 38 - 39, included healthcare, environment, health

Materials and methods removed

Line 131 - 136, justification of table 1

Line 351 - 356, justification of table 2

Figure 4, now figure 2 is the intellectual property of the authors.

Round 2

Reviewer 1 Report

1. Abstract: this is too long. kindly shorten the sentences. 

2. The conclusion: is too long. It is suggested to include subtopics towards the end of the manuscript to address info in the conclusion. 

3. in conclusion: only address or highlight the outcomes as per addressed in the objectives. 

Author Response

The abstract was shortened.

The conclusion was shortened.

All the grammar suggestions were made.

In the conclusion, only addressed outcomes that are in the objectives.